# Link-State Aware Hybrid Routing in the Terrestrial–Satellite Integrated Network [note 1]

**DOI:** 10.3390/s22239124

**Published:** 2022-11-24

**Authors:** Huihui Xu, Zhangsong Shi, Mingliu Liu, Ning Zhang, Yanjun Yan, Guangjie Han

**Affiliations:** 1School of Weaponry Engineering, Naval University of Engineering, Wuhan 430000, China; 2Electronic Information School, Wuhan University and with the State Grid Hubei Electric Power Research Institute, Wuhan 430072, China; 3The 91458 Troops, Sanya 572021, China; 4Department of Information and Communication Systems, Hohai University, Changzhou 213022, China

**Keywords:** terrestrial–satellite integrated networks, hybrid routing, space–time topology, link-state aware

## Abstract

In this paper, we study data transmission in the Terrestrial–Satellite Integrated Network (TSIN), where terrestrial networks and satellites are combined together to provide seamless global network services for ground users. However, efficiency of the data transmission is limited by the time-varying inter-satellite link connection and intermittent terrestrial–satellite link connection. Therefore, we propose a link-state aware hybrid routing algorithm, which selects the integrated data transmission path adaptively in this paper. First of all, a space–time topology model is constructed to represent the dynamic link connections in TSIN. Thus, the transmission delay can be analyzed accordingly, and the data transmission problem can then be formulated. To balance the effectiveness and accuracy of searching a hybrid path, we carefully discuss the optimization of space–time topology updating, and propose an inter-satellite link selection algorithm. For the terrestrial–satellite link in hybrid routing, the data transmission problem is transformed into a weighted bipartite graph matching problem and solved with a Kuhn–Munkres-based link selection algorithm. To verify the effectiveness of our proposed routing algorithm, extensive simulations are conducted based on a realistic Hongyun constellation project. Results show that the network performance is improved with respect to data transmission delay, packet loss rate, and throughput.

## 1. Introduction

In the past few decades, the rapid development of terrestrial wireless communication has triggered increasing demand for high broadband and massive access applications, such as augmented reality, virtual reality, and high definition video, which in turn, raised emergent requirements on achieving massive connectivity and high capacity in future communication systems [1,2]. However, limited by the constrained power storage or insufficient base station deployment, the network coverage in rural or harsh areas is quite low. Due to the wide coverage, broad operating spectrum, and ultra-dense topology, Low Earth Orbit (LEO) satellite networks have been identified as the most cost-affordable technology to meet the terrestrial coverage requirements [3,4]. Several companies have announced their plans to launch thousands of LEO satellites, including SpaceX, Telesat, Kuiper, and Hongyun [5,6], where ultra-dense constellations are normally deployed to support high-capacity communication services [7,8]. In order to accommodate new bandwidth-intensive applications and the very significant upsurge in the number of user equipment (UEs) devices beyond 5G and 6G, inter-connectivity and resource sharing between the communication networks on the ground, at sea, in the air, and in space will be required [9]. As a combination of satellite networks and terrestrial networks, TSIN has become a promising solution for providing seamless global network services, and mitigating network congestion, especially when the traffic exceeds capacity of the terrestrial links [10].

In most planned TSINs, satellite systems are operated with dedicated ground stations for data transmission, which means that with the limited number of ground infrastructures, the satellite communication system would work in a store-and-forward mode for data exchange [11]. Hence, the increasing data volume could cause high transmission delays or serious traffic congestion. With the development of satellite communication techniques, Terrestrial–Satellite gateway Equipments (TSEs) can communicate with satellites directly and be regarded as satellite access points in data transmission [12]. Given a wide distribution of smart TSEs in the near future, the hybrid routing in TSIN becomes a promising solution to mitigate the traffic burden of satellite or terrestrial networks, which however, is quite challenging due to the following reasons. First, the TSE-satellite link connections are intermittent and dynamic; thus, the throughput of the data transmission will be reduced. Second, due to the rapid satellite movement, network topology needs to be updated frequently. If the time interval of topology updating is too large, the link delay will be inaccurate, which results in suboptimal path planning, while the short value of the time interval will lead to an increase of network overhead. Finally, the unbalanced transmission resources of TSIN could cause local communication congestion, and finally degrade the transmission efficiency.

With the increasing demand of seamless global communication services for UTs, the TSIN routing strategy has attracted more and more attention in recent years. Basically, the routing methods can be classified into relay routing [13,14] and extended terrestrial routing [15,16,17]. In the relay routing method, satellites act as relay forwarding nodes, which are interconnected through ground stations. Therefore, short delay can hardly be guaranteed because of the data forwarding dependence on the limited number of ground stations. For the extended terrestrial routing mode, terrestrial routing algorithms are extended to the satellite networks. However, due to the rapid movement of satellites, the communication link connection should be updated frequently. To the best of our knowledge, the combination of terrestrial and satellite networks is still an open issue, and the hybrid routing design problems jointly considering the practical issues mentioned above are still missing, thus motivating our investigation in this paper.

In this work, a link-state aware Terrestrial-Satellite Hybrid Routing (TSHR) algorithm is proposed to find a data transmission path that spans both inter-satellite and terrestrial links for TSIN. Firstly, a space–time topology graph is proposed to analyze the dynamic relationship of network resources in the time and space domain. Then, under the communication link capacity and connection constraints, the hybrid routing problem is modeled as an integer linear programming problem. The inter-satellite link and terrestrial–satellite link connection are time-varying and intermittent in TSIN, which makes the optimal data transmission path planning a challenging issue. By exploiting the space–time topology structure, we propose a Time interval optimization-based Satellite Routing (TSR) algorithm to resolve the optimum inter-satellite link connection problem. Based on the TSR algorithm, with an objective of minimizing the end-to-end delay for most reliable connections, the hybrid TSIN routing problem is formulated and transformed into a weighted bipartite graph matching problem for solution search. Then, the TSHR algorithm is proposed to find the optimal TSE-satellite communication link connection. The contributions of this paper are summarized as below:
Link-state aware hybrid routing: Although existing works have already studied the communication link planning according to the network topology information, the intermittent and dynamic nature of the TSE–satellite link selection is still missing. In our work, we propose a link-state aware hybrid routing scheme for data transmission in the dynamic TSIN. The design is under a practical communication scenario that the link connections in satellites and resources are time-varying. By deriving the delay of both inter-satellite and TSE–satellite links, we formulate a hybrid routing problem aimed at minimizing the data transmission delay by optimally selecting the inter-satellite and TSE–satellite links.Characterization on TSIN topology: The updating time of the topology structure is assumed to be a fixed time interval in existing models, which introduces significant difficulties in balancing the network overhead and accurate transmission delay. To tackle this issue, we propose a space–time topology graph to represent the data transmission process in dynamic TSIN, where the link resources in both the spatial and temporal dimensions are quantified. In particular, we derive the optimal time updating interval of the topology relationship, with which the effectiveness and accuracy of obtaining the hybrid path can be balanced.Decomposition solution: The inter-satellite link and terrestrial–satellite link connection are time-varying and intermittent in TSIN, which makes the optimal data transmission path planning a challenging issue. By exploiting the space–time topology structure, we propose a satellite link selection algorithm to resolve the optimum inter-satellite link connection problem. Then, a hybrid routing algorithm based on a bipartite graph is proposed to determine the optimal TSE–satellite communication link connection, which can solve the subproblems with smaller sizes.

The rest of this paper is organized as follows. Section 2 reviews the related works, then the system model is described together with a space–time topology model, based on which the delay minimization problem is formulated in Section 3. In Section 4, the TSIN hybrid routing scheme is proposed. Finally, we provide our simulation results in Section 5 and conclude this paper in Section 6.

## 2. Related Work

In this section, recent studies on the TSIN routing method are reviewed, which can be classified into two types: the satellite routing schemes and the hybrid terrestrial–satellite routing schemes.

### 2.1. Satellite Routing Schemes

Some researchers presumed that propagation delay is the dominating factor in satellite communication, and a number of pioneering satellite routing mechanisms have been proposed to find the minimum propagation delay paths with minimal hop count for communication. According to the periodic motion of satellites, Hong et al. [18] proposed a routing strategy based on a finite state to calculate the global optimal path of each state. However, the routing table for each slot is calculated on ground and stored onboard in advance, which can not avoid the link interruption caused by the time-varying topology. In [19], based on a dynamic virtual topology, a priority-based adaptive routing algorithm is proposed to find the minimum distance path by the Dijkstra algorithm. Ekici et al. [20] proposed a routing strategy based on geographic distance with minimized propagation delay between satellites. Li et al. [21] proposed a Netgrid-based Shortest path Routing (NSR) algorithm to find the optimal path from the source netgrid to any other reachable netgrids. However, the search for the distance path would cause congestion in some satellites, which leads to not only unfair distribution of the network traffic, but also higher queuing delay at some satellites. Therefore, in [22,23], agent-based dynamic routing schemes are presented to collect queuing delays of all neighbor links and deliver them to all satellites. In these schemes, the satellite operation period can be divided into multiple time intervals, queuing delay is collected at the beginning of each interval, and the Dijkstra Shortest-Path (DSP) algorithm is used to calculate the shortest path. Since the gathering of global link takes longer distances, the signaling exchange time would increase. Thus, it occupies more channel resources, which is reflected in a larger communication overhead.

To cope with this issue, a distributed routing strategy is proposed, in which the routing decisions are made independently for each packet. For example, to avoid traffic congestion, the Explicit Load Balancing (ELB) routing method [24] focuses on making a choice for the next best hop, in which the congestion status is exchanged among neighboring satellites. Based on current outgoing links, the Traffic-Light-based Routing (TLR) algorithm [25] is proposed to achieve load balance by deriving the queue length. In TLR, the ISL congestion is predicted to re-calculate routing. Another proactive distributed routing method is the selective split load balancing strategy (SSLB) [26], in which the traffic flow of congestion nodes is diverted to neighboring nodes. However, all ISLs are assumed to have equal length, it can hardly reflect the time-varying feature of propagation delay in real satellite networks.

### 2.2. Hybrid Terrestrial–Satellite Routing Schemes

To transmit data via terrestrial links and satellite links, the Broadband Satellite Multimedia (BSM) system based on gateway is proposed [13]; the satellites are taken as a relay node of the terrestrial network through gateways. To reduce link failures or congestion, a Hybrid Network Management (HNM) method based on smart antenna [14] is proposed, in which antenna arrays at the backhaul nodes are deployed to enable topology reconfiguration through ground station–satellite links. However, limited by the deployment of ground stations, the communication capabilities of TSINs are greatly restricted. To realize unified routing management of TSINs, several extended terrestrial routing protocols are adopted and modified for the satellite networks, such as the modified Open Shortest Path First (OSPF) [17] and Border Gateway Protocol (BGP) [15], in which state information of all inter-satellite links is collected to calculate routing table periodically, which will result in significant computational and signaling complexity for high dynamic satellite networks. In recent years, software defined network (SDN)-based routing algorithms have been proposed to centrally manage network routing [27,28,29], then greedy algorithm is used to obtain the global optimal path based on the real-time link state. However, the time-varying topology of the network will bring frequent path updates and increase the computational overhead.

Graph theory has been proposed and drawn a significant amount of research interests in predictable dynamic networks [30,31,32,33,34,35,36,37,38]. Ivanov et al. proposed graph-based resource allocation for integrated space and terrestrial communications in [30], where the inter-connectivity and coexistence of various terrestrial and non-terrestrial networks are described via a hypergraph and its attributes. To model the network topology, graph-based deep learning has achieved good performance in a series of problems in communication networks. Jiang [31] reviews the research using different graph-based deep learning models, e.g., graph convolutional and graph attention networks, in various problems from different types of communication networks. The applicability of the graph-based deep learning in the construction of efficient routing protocols in realistic scenarios is analyzed in [32,33]. In [32], by exploiting the graph neural network’s topology extraction capabilities, a graph neural network-based learning approach for routing in a satellite network is proposed. However, learning-based dynamic routing approaches usually have poor generalization ability for different topologies and incomplete state information, which will degrade their performance in a dynamic environment. To handle the time-varying topology, frequent link handover and imbalanced communication load, GRouting algorithm combining graph neural networks with deep reinforcement learning is proposed to dynamically find the optimal routing paths for the LEO satellite. To achieve efficient observation data transmission, recent works [34,35,36] design a communication link planning algorithm according to the network topology information. For example, according to the greedy forwarding strategy, Lv et al. [34] proposed a hyperbolic geometry-based routing strategy to decrease storage overhead. Furthermore, a service-aware communication link planning algorithm [37,38] is proposed to improve network performance. However, a careful study with the consideration of the frequent changes of inter-satellite link and TSE-satellite link selection is missing.

Table 1 summarizes the different routing algorithm features. The routing algorithms based on virtual topology and node simplify the mobility of satellite networks in a periodic sequence. To avoid link congestion, load balancing routing algorithms update the routing path according to the congestion state step-by-step. The relay routing and extended terrestrial routing mainly focus on data forwarding around the ground station. Due to the limited number of ground stations, the short delay can hardly be guaranteed. In graph-based routing, the dynamic topology is modeled as a time-involved graph. Even though global information can be utilized to optimize the routing paths, the selected terrestrial-satellite link selection could disappear due to the frequent moving satellite. To this end, a dynamic analysis of the time-varying links is fundamental for enhancing the throughput. We exploit the hybrid routing algorithm in [39] to provide seamless integration of the satellite network with terrestrial networks, in which transmission paths can be selected adaptively according to the traffic demands of user terminals. Instead of setting a fixed time interval, an optimal topology updating interval is derived in this paper, through which effectiveness of the hybrid routing path is further improved.

## 3. System Model

In this section, we first introduce the data transmission architecture of TSIN, then we propose the space–time topology model of TSINs, and, constrained by the intermittent connectivity, the hybrid routing problem will be formulated.

### 3.1. Scenario Description

Although some terrestrial networks have been proposed to deal with the urgent requirement for massive user connections, due to the increasingly scarce resources, limited coverage range, and severe damage caused by force majeure factors, the terrestrial networks will face local congestion and failure, which affects the overall network transmission performance. Thus, the proposed solution is mainly intended to address the mobile communications scenario, which would increase the communication capacity and guarantee network connectivity in natural disasters. In this way, the data from the terrestrial terminal can be transmitted to the satellite networks conveniently and seamlessly through the TSE; therefore, reducing the burden on terrestrial communication networks. A general data transmission architecture of TSIN is presented in Figure 1, which comprises two main segments: satellite and ground. These two segments can work independently or inter-operationally, by integrating heterogeneous networks among the two segments, it is easy to build a hierarchical broadband wireless network. The satellite network is composed of orbit satellites $S=\{s_{1},s_{2},\ldots ,s_{M}\}$ and constellations as well as their corresponding terrestrial infrastructures (e.g., TSEs $G=\{g_{1},g_{2},\ldots ,g_{N}\}$). The ground network mainly consists of terrestrial communication systems, such as cellular networks, wireless local area networks, and so on. The ground network is able to provide high data rates to users, but the network coverage in rural and remote areas is limited. Satellite networks can provide global coverage on the earth but have long propagation latencies. Although ground networks have the lowest transmission delay, they are vulnerable to natural disasters or artificial infrastructure damage. Mobile user terminals $UT=\{UT_{1},UT_{2},\ldots ,UT_{L}\}$ are assumed to be randomly distributed, and the session data can be transmitted from ground to space only when they are covered by specific satellites. TSEs play an important role in the process of traffic delivery from the ground network to the satellite network. It is noticed that different selections of TSEs would produce varying transmission delay. Thus, selecting the appropriate TSEs can enhance the network performance. As shown in Figure 1, there are three kinds of transmission links in TSINs:-Inter-satellite link (ISL): the link between satellites, the distance of which would change rapidly with the highly moving constellations.-TSE-satellite link (TSL): the link between a TSE and a satellite. Since a satellite can cover different TSEs for a different time, the TSL is time-varying.-Inter-terrestrial link (ITL): the link between terrestrial nodes. As the terrestrial network service equipments are usually static, the ITL distance is invariant.

During a hybrid network communication session, the data could be transmitted through ITLs, TSLs, and ISLs at the same time. Generally, the TSL can only be established by one satellite at one time, due to the rapid movement of satellites, the communication link between one satellite and one TSE would be handed over to other satellite. Thus, the communication link of satellite-TSE is time-varying. Although the space segment is designed to provide continuous coverage to TSEs, a TSE would be covered by one satellite at one time, and it is a dynamic spatial coverage relation, i.e., a TSE needs to be served by several satellites during a communication session. Due to the time-varying TSE–satellite link, one of the most critical issues in designing a hybrid routing mechanism is to schedule the transmitted data and link resources.

### 3.2. Space–Time Topology Model

To characterize TSIN resources and capture the dynamic topology evolution, according to the link information divided by discrete time intervals, a space–time topology graph is proposed to represent the relationship of link resources in the time and space domain. In particular, the path plan time is divided into consecutive time intervals indexed by t∈{1,…,D}. During each time interval Δτ, the network topology of TSIN is considered as stable or quasi-static since the span of the time interval is short. Therefore, a static graph can be adopted to describe the interactions of nodes in each time interval. We defined this static graph as a snapshot of the topological evolution. Through a sequence of static graphs, the time-varying topology can be presented as a space–time topology graph as follows: the space–time topology graph G(t)=(V,E(t)) is composed of *D* layers, corresponding to the network topology of *D* time intervals shown in Figure 2. The vertices of *G* corresponding to the set of UTs, LEO satellites, and TSEs for each time interval. The vertices set is denoted as $V=V_{UT}\cup V_{s}\cup V_{g}$, where $V_{UT}=\left\{UT_{i}^{t}|UT_{i}\in UT,1\leq t\leq D\right\}$, $V_{s}=\left\{s_{i}^{t}|s_{i}\in S,1\leq t\leq D\right\}$, $V_{g}=\left\{g_{i}^{t}|g_{i}\in G,1\leq t\leq D\right\}$. The key notations are summarized in Table 2.

Since the graph representing the connectivity between these nodes varies with time, E(t) is a time varying set of links between these nodes, which consists of a spatial link and a temporal link. For the sake of simplification of the model, the link between UT and TSE includes the potential terrestrial paths between TSEs. Wherein, the set of spatial links is denoted as $E_{l}\left(t\right)=E_{gu}\left(t\right)\cup E_{sg}\left(t\right)\cup E_{ss}\left(t\right)$, and the links in $E_{gu}\left(t\right)$ are shown as black lines in Figure 2, which represent the communication opportunities for UTs to TSE in each time interval, i.e., $E_{gu}\left(t\right)=\left\{\left(UT_{i}^{t},g_{i}^{t}\right)\right|1\leq t\leq D,$$g_{i}$ is the access TSE of $UT_{i}$ in the *t*-th time interval}. Similarly, the links in $E_{ss}\left(t\right)$ and $E_{sg}\left(t\right)$ represent the communication opportunities for satellite to satellite and satellite to TSE in each time interval, where $E_{ss}\left(t\right)=\left\{\left(s_{i}^{t},s_{j}^{t}\right)\right|1\leq t\leq D,$$s_{j}^{t}$ is the neighbor satellite of $s_{i}^{t}$ in the *t*-th time interval} and $E_{sg}\left(t\right)=\left\{\left(s_{i}^{t},g_{i}^{t}\right)\right|1\leq t\leq D,$$g_{i}$ is in the coverage of $s_{i}$ in the *t*-th time interval}. Since nodes $v_{i}^{t},t=1,\ldots ,D$ in G(t) correspond to the single node $v_{i}$ in the network, $v_{i}^{t}$ and $v_{i}^{t+1}$ can be connected by a temporal link $v_{i}^{t}\to v_{i}^{t+1}$ (horizontal links in Figure 2), which represents node $v_{i}$ storing messages in the *t*-th time interval, i.e., traversing a temporal link denotes “carrying” a message by a satellite. Intuitively, the data transmission on dynamic links can imagine a “band” (of message transmission duration) going from left to right as we travel from source to destination. Occasionally this band has to “slide” toward the right when there is no link available for transmission immediately to the chosen next hop node. For example, in Figure 2, after receiving the message from $s_{1}^{1}$, $s_{2}^{1}$ could not forward it immediately to $s_{3}^{1}$, but had to carry it for some time until the link $\left(s_{2}^{2},s_{3}^{2}\right)$ became active. Since the ground stations always have massive storage, we only focus on the limited buffer size for satellites. The set of temporal links is denoted as $E_{b}\left(t\right)=\left\{\left(v_{i}^{t},v_{i}^{t+1}\right)|v_{i}^{t}\in V_{s}\cup V_{g},1\leq t\leq D-1\right\}$, $E\left(t\right)=E_{l}\left(t\right)\cup E_{b}\left(t\right)$. By defining the space–time graph G, any communication operation in the time-evolving network can be simulated on this directed graph. This guarantees that the packet can be delivered between any two nodes in the network over the period *D*.

Since the antenna or laser terminal of the TSL and TSE requires more complex capture, tracking, and aiming mechanisms, it is difficult to install multiple antennas at the same time on the satellite platform. Therefore, by using the rotation-limited beam or fixed beam, a satellite can set up ISLs with four neighbors at most, with two in its own orbit and the other two in the neighboring orbits [24,25], so as to realize the optimal constellation design, such as Iridium, SpaceX, Telesat, Kuiper, and Hongyun [5,6]. It can also establish a TSL with the TSEs in its coverage, where all TSEs are represented by a single ground antenna for simplicity. In this way, the four ISLs would be established at the same time, while one TSL is established by one satellite at one time. Considering the intermittent connectivity of ISLs and TSLs, the hybrid routing mechanism of TSIN is modeled as an integer linear programming problem, aimed at link selection by minimizing the network delay. The network delay usually includes processing delay, queuing delay, transmission delay, and propagation delay. Wherein, the processing delay is related to hardware performance which is almost negligible. The link delay of graph G(t) represents the sum of data propagation delay $T_{p}\left(v_{i}^{t},v_{j}^{t}\right)$, queuing delay $T_{q}\left(v_{i}^{t},v_{j}^{t}\right)$, and transmission delay $T_{w}\left(v_{i}^{t},v_{j}^{t}\right)$ in a time interval, which can be expressed as:(1)$$\begin{array}{cc} T(v_{i}^{t},v_{j}^{t}) & =T_{p}\left(v_{i}^{t},v_{j}^{t}\right)+T_{q}\left(v_{i}^{t},v_{j}^{t}\right)+T_{w}\left(v_{i}^{t},v_{j}^{t}\right)\\ \; & =\frac{d(v_{i}^{t},v_{j}^{t})}{v_{P}}+\frac{\sum_{k}q_{i}^{t}\left(k\right)}{r(v_{i}^{t},v_{j}^{t})}+\frac{\sum_{k}\vartheta (v_{i}^{t},v_{j}^{t})}{r(v_{i}^{t},v_{j}^{t})}, \end{array}$$
where $d(v_{i}^{t},v_{j}^{t})$ is the distance of link $\left(v_{i}^{t},v_{j}^{t}\right)$. $v_{P}$ denotes the transmission speed of radio signal, $r(v_{i}^{t},v_{j}^{t})$ denotes the capacity of link $\left(v_{i}^{t},v_{j}^{t}\right)$ (in bit/s), $q_{i}^{t}\left(k\right)$ is the data size in queue of $v_{i}^{}$ for $UT_{k}$ data during the *t*-th time interval (in bit), $\vartheta (v_{i}^{t},v_{j}^{t})$ is the flow on link $\left(v_{i}^{t},v_{j}^{t}\right)$ (in bit).

### 3.3. Problem Formulation

Constrained by the link intermittent connectivity, ISLs and TSLs have time-varying capacity and connection. In this case, there are three basic types of constraints, i.e., link connection constrain, data flow balance constraints, and communication velocity constraints.

(1) Link Connection Constraints: To model the contention caused by dynamic satellites, a set of boolean variables is introduced.
(2)$$\omega \left(v_{i}^{t},v_{j}^{t}\right)=\{0,1\},\forall \left(v_{i}^{t},v_{j}^{t}\right)\in E_{sg}\cup E_{ss},$$
which takes the value of 1 if link $\left(v_{i}^{t},v_{i}^{t}\right)$ is active, and otherwise.

Due to the limitation of antenna pointing, most LEO constellation systems can establish inter-satellite links with four satellites around them, such as Iridium and Hongyun, each satellite has two neighbors on the same orbit and two neighbors on the adjacent orbit, i.e.,
(3)$$C1:\sum_{\left(v_{i}^{t},v_{j}^{t}\right)\in E_{ss}}\omega (v_{i}^{t},v_{j}^{t})\leq 4.$$

Since the ISL is bidirectional, $\omega (v_{i}^{t},v_{j}^{t})$ need to satisfy:(4)$$C2:\omega \left(v_{i}^{t},v_{j}^{t}\right)=\omega \left(v_{j}^{t},v_{i}^{t}\right),\forall \left(v_{i}^{t},v_{i}^{t}\right)\in E_{ss}.$$

Meanwhile, the TSE can only be established by one satellite at one time, i.e.,
(5)$$C3:\sum_{\left(v_{i}^{t},v_{j}^{t}\right)\in E_{sg}}\omega (v_{i}^{t},v_{j}^{t})\leq 1,\forall v_{i}^{t}\in V_{s},v_{j}^{t}\in V_{g}.$$

(2) Data Flow Balance Constraints: to balance the change in the storage occupancy of a satellite against the incoming data, we define $b_{i}^{t}\left(k\right)$ as the data volume of mission flow $UT_{k}$ in the storage of node $v_{i}$ at the end of the time interval *t*; thus, the amount of all incoming flows and all outgoing flows at a node must satisfy the flow conservation. The data flow balance constraint can be expressed as:(6)$$\begin{array}{c} C4:\sum_{\left(v_{i}^{t},v_{j}^{t}\right)\in E_{l}}\vartheta \left(v_{i}^{t},v_{j}^{t},k\right)+b_{i}^{t}\left(k\right)=\sum_{\left(v_{i}^{t},v_{j}^{t}\right)\in E_{l}}\vartheta \left(v_{j}^{t},v_{i}^{t},k\right)+b_{i}^{t-1}\left(k\right). \end{array}$$

Furthermore, the aggregate data in a link should not exceed the maximum amount of data that can be transmitted by this link, thus we have:(7)$$C5:\sum_{k}\vartheta \left(v_{i}^{t},v_{j}^{t},k\right)\leq C\left(v_{i}^{t},v_{j}^{t}\right).$$

(3) Communication Velocity Constraints: the transmission velocity needs to satisfy the minimum rate requirement of the data transmission for UT, thus we have:(8)$$C6:r\left(v_{i}^{t},v_{j}^{t}\right)\geq r_{\min}.$$

Therefore, combining the restrictions of link connection, data flow balance, and communication velocity, the hybrid routing problem can be formulated to minimize the end-to-end delay and be expressed as:(9)$$\begin{array}{c} \begin{array}{cc} \min_{\vartheta ,\omega ,b} & \sum_{\left(v_{i}^{t},v_{j}^{t}\right)\in E_{l}}\left(\frac{d(v_{i}^{t},v_{j}^{t})}{v_{P}}+\frac{\sum_{k}q_{i}^{t}\left(k\right)}{r(v_{i}^{t},v_{j}^{t})}+\frac{\sum_{k}\vartheta (v_{i}^{t},v_{j}^{t},k)}{r(v_{i}^{t},v_{j}^{t})}\right)\omega \left(v_{i}^{t},v_{j}^{t}\right) \end{array}\\ \begin{array}{cc} s.t.\;C1:\sum_{\left(v_{i}^{t},v_{j}^{t}\right)\in E_{ss}}\omega (v_{i}^{t},v_{j}^{t})\leq 4, & \end{array}\\ C2:\omega \left(v_{i}^{t},v_{j}^{t}\right)=\omega \left(v_{j}^{t},v_{i}^{t}\right),\forall \left(v_{i}^{t},v_{i}^{t}\right)\in E_{ss},\\ C3:\sum_{\left(v_{i}^{t},v_{j}^{t}\right)\in E_{sg}}\omega (v_{i}^{t},v_{j}^{t})\leq 1,\forall v_{i}^{t}\in V_{s},v_{j}^{t}\in V_{g},\\ C4:\sum_{\left(v_{i}^{t},v_{j}^{t}\right)\in E_{l}}\vartheta \left(v_{i}^{t},v_{j}^{t},k\right)+b_{i}^{t}\left(k\right)=\sum_{\left(v_{i}^{t},v_{j}^{t}\right)\in E_{l}}\vartheta \left(v_{j}^{t},v_{i}^{t},k\right)+b_{i}^{t-1}\left(k\right),\\ C5:\sum_{k}\vartheta \left(v_{i}^{t},v_{j}^{t},k\right)\leq C\left(v_{i}^{t},v_{j}^{t}\right),\\ C6:r\left(v_{i}^{t},v_{j}^{t}\right)\geq r_{\min}. \end{array}$$

Problem (Equation 9) is a mixed-integer nonlinear program optimization problem that is challenging to solve. Furthermore, in the formulated optimization problem, ω plays a key role in the solution since it determines which ISLs are available to transmit communication data. Considering both of the two above difficulties, to obtain the optimal transmission paths, we decompose the hybrid routing Problem (Equation 9) into two subproblems, i.e., the ISL connection subproblem and weighted bipartite graph matching subproblem.

## 4. Link-State Aware Hybrid Routing Scheme

In this section, to balance the effectiveness and accuracy of obtaining the hybrid path, the optimal time updating interval of the topology relationship is firstly discussed. Then, the hybrid routing algorithm is proposed to obtain the data transmission path with the minimum delay.

### 4.1. Time Interval Optimization

In LEO constellation, due to the dynamic and time-varying feature of satellite networks, the ISL in different orbits will be interrupted. When the angular velocity between satellites in the polar region becomes large, the interplane ISL would be disconnected. Since the space–time topology of TSIN can be considered as fixed during each time interval, the duration of the time interval will affect the routing performance caused by interplane ISL variation. To obtain the optimal inter-satellite link connection, we need to find the optimum time interval. Towards this end, by analyzing the distance of dynamic ISL, we find a relationship function between the time interval and the satellite constellation parameters.

We assume that they are *N* polar orbits in the satellite network, and each of them contains *M* satellites. Therefore, the angular distance between two adjacent satellites in the same orbit is $\frac{360^{\circ }}{M}$. The distance of intraplane ISL is *L*, which can be expressed as:(10)$$L=r\times \sqrt{2(1-\cos\left(360^{\circ }/M\right))},$$
where *r* is the radius of the orbit.

The geometry expression of a satellite is illustrated in Figure 3, where $\phi_{s}\left(t\right)$ denotes the declination of a satellite in orbit, $\lambda_{s}\left(t\right)$ is the right ascension of a satellite in orbit, and the inclination angle of the orbital plane is θ. Then, according to the satellite’s orbital parameters, the location of the satellite on orbital sphere at time *t* can be obtained as:(11)$$\left\{\begin{array}{c} \phi_{s}\left(t\right)=\arcsin\left(\sin\theta \times \sin u\left(t\right)\right),\\ \lambda_{s}\left(t\right)=\Omega_{0}+\arctan2\left(\frac{\sin\phi_{s}\left(t\right)}{\tan\theta },\cos u\left(t\right)\right)-\omega_{e}\times t,\\ u(t)=w\times t+\gamma , \end{array}\right.$$
where $\Omega_{0}$ denotes the position of the ascending intersection, and the angular velocity of the satellite rotating around the earth is *w*. γ denotes the initial phase angle of the satellite, and $\omega_{e}$ is the angular velocity of Earth’s rotation.

Based on the celestial sphere coordinate system, the coordinates of a satellite on an orbital sphere at time *t* is expressed as:(12)$$\left\{\begin{array}{c} x\left(t\right)=r\times \cos(\phi_{s}\left(t\right)\times \cos\left(\lambda_{s}\left(t\right)\right),\\ y\left(t\right)=r\times \cos(\phi_{s}\left(t\right)\times \sin\left(\lambda_{s}\left(t\right)\right),\\ z\left(t\right)=r\times \sin(\phi_{s}\left(t\right)). \end{array}\right.$$

The time-varying distance of the interplane link between satellite $s_{i}$ and $s_{j}$ is $d(s_{i}^{t},s_{j}^{t})$, which is derived as:(13)$$d\left(s_{i}^{t},s_{j}^{t}\right)=\sqrt{{\left(x_{i}\left(t\right)-x_{j}\left(t\right)\right)}^{2}+{\left(y_{i}\left(t\right)-y_{j}\left(t\right)\right)}^{2}+{\left(z_{i}\left(t\right)-z_{j}\left(t\right)\right)}^{2}}.$$

Combining (Equation 12) and (Equation 13), the ISLs length is derived as a time-dependent parameter, which can be expressed as:(14)$$d\left(s_{i}^{t},s_{j}^{t}\right)=r\sqrt{f(s_{i}^{t},s_{j}^{t})},$$
(15)$$f\left(s_{i}^{t},s_{j}^{t}\right)=2\left[1-\cos\phi_{i}\left(t\right)\times \cos\phi_{j}\left(t\right)\times \sin\left(\lambda_{i}\left(t\right)+\lambda_{j}\left(t\right)\right)-\sin\phi_{i}\left(t\right)\times \sin\phi_{j}\left(t\right)\right].$$

To explore the time-varying features of ISL, the ISL distance of the Iridium and Hongyun constellation is compared. As shown in Table 3, the simulated satellite orbit parameters and the propagation delay of ISLs are presented. The comparison results show that the intraplane ISL length increases with the increasing of orbital height, and decreases with more satellites distributed in the orbit plane. Similarly, the interplane ISL distance increases with the increasing of orbital height and inclination, while it decreases with the satellites number in an orbit plane. The change of interplane ISL lengths are simulated with Iridium and Hongyun project, and as shown in Figure 4, due to the difference in satellite constellation parameters, the ISL distance varies periodically in the time domain, and the change of Iridium ranges from 1680 km to 4370 km, which is more than two times wider than that of Hongyun.

Based on the above analysis, the distance of interplane ISL is time-varying, and the orbiting movements of satellites also lead to dynamic distances of TSL. Meanwhile, the achievable data rate of ISL is inversely proportion to the square of distance [38]. In other words, ISLs have time-varying link capacity. If the duration of Δτ is too large, the transmission delay of the link will be inaccurate, which results in suboptimal path planning, while the short value of Δτ will waste the computing resources. To this end, based on the dynamic characteristics of ISL distance, the relationship function between time interval and satellite constellation parameters can be derived.

Assuming that the shortest path *U* between satellites in satellite network $G_{s}\left(V_{s},E_{ss}\right)$ is composed of *N* links, the propagation delay of the shortest path *U* is $\sum_{\left(s_{i}^{t},s_{j}^{t}\right)\in E_{ss}}^{N}T_{p}\left(s_{i}^{t},s_{j}^{t}\right)$. Since the ISL distance is continuous over time, the satellite network $G_{s}\left(V_{s},E_{ss}\right)$ will be static in a time interval Δτ, i.e., the neighbors of each satellite in Δτ can not change. To obtain the optimum time interval, the delay corresponding to ISL distance variations are adopted as the reference for the Δτ setting, so that the path accuracy and computing resources can be guaranteed. Denoting the maximum delay corresponding to ISL length variation as λ, all propagation delays after Δτ need to satisfy:(16)$$|\sum_{\left(s_{i}^{t},s_{j}^{t}\right)\in E_{ss}}^{N}T_{p}\left(s_{i}^{t+\Delta \tau },s_{j}^{t+\Delta \tau }\right)-\sum_{\left(s_{i}^{t},s_{j}^{t}\right)\in E_{ss}}^{N}T_{p}\left(s_{i}^{t},s_{j}^{t}\right)|\leq \lambda .$$

Let $T_{\max}\left(t\right)$ denotes the maximum interplane ISL propagation delay of the shortest path *U*, then we have
(17)$$\sum_{\left(s_{i}^{t},s_{j}^{t}\right)\in E_{ss}}^{N}T_{p}\left(s_{i}^{t},s_{j}^{t}\right)<N\times T_{\max}.$$

Denoting $\Delta d_{\max}\left(s_{i}^{\Delta \tau },s_{j}^{\Delta \tau }\right)$ as the maximum distance change of ISL in a Δτ, (Equation 17) can be transformed as
(18)$$|\sum_{\left(s_{i}^{t},s_{j}^{t}\right)\in E_{ss}}^{N}T_{p}\left(s_{i}^{t+\Delta \tau },s_{j}^{t+\Delta \tau }\right)-\sum_{\left(s_{i}^{t},s_{j}^{t}\right)\in E_{ss}}^{N}T_{p}\left(s_{i}^{t},s_{j}^{t}\right)|\leq \frac{N\times \Delta d_{\max}\left(s_{i}^{\Delta \tau },s_{j}^{\Delta \tau }\right)}{v_{p}}.$$

Based on (Equation 16) and (Equation 18), $\Delta d_{\max}\left(s_{i}^{\Delta \tau },s_{j}^{\Delta \tau }\right)$ can be formulated as
(19)$$\Delta d_{\max}\left(s_{i}^{\Delta \tau },s_{j}^{\Delta \tau }\right)\leq \frac{\lambda \times v_{p}}{N}.$$

Since $\Delta d_{\max}\left(s_{i}^{\Delta \tau },s_{j}^{\Delta \tau }\right)$ is a continuous function of Δτ, when $\Delta \tau =\Delta \tau_{\max}$, we have $\Delta d_{\max}\left(s_{i}^{\Delta \tau },s_{j}^{\Delta \tau }\right)=\frac{\lambda \times v_{p}}{N}$. According to (Equation 14) and (Equation 15), the relationship function between the maximum Δτ and satellite constellation parameters is expressed as
(20)$$|\sqrt{f(s_{i}^{t+\Delta \tau },s_{j}^{t+\Delta \tau })}-\sqrt{f(s_{i}^{t},s_{j}^{t})}|=\frac{\lambda \times v_{p}}{r\times N}.$$

### 4.2. Link-State Aware Hybrid Routing Algorithm

Although the *b* and ϑ are coupled through the capacity constraint, they can be decoupled by employing the special characteristic of our problem. Intuitively, with the given inter-satellite contact connection, minimizing the transmission delay for UTs gives the satellites the opportunity to delivery as much of their data as possible, which facilitates their mission completion velocity. Therefore, based on the time interval optimization, the ISL connection problem can be solved by minimizing the transmission delay for each satellite by optimally allocating the link resource of each satellite. When the optimal ISL connection planning during each time interval is obtained, a hybrid routing algorithm based on a bipartite graph is proposed to find the optimal TSL connection. To this end, the overall optimization of the hybrid routing problem (Equation 9) can be achieved by successively optimizing the ISL and TSL connection planning.

#### 4.2.1. Time Interval Optimization-Based ISL Selection Algorithm

To reduce the search space, according to the optimized time interval, the ISL selection algorithm is proposed. Assuming that data of multiple UTs can be transmitted independently, the routing mechanism of communication data for each UT can be designed independently, i.e., for the communication data of $UT_{k}$, the route between candidate satellites associated with TSE can be determined. Since a TSE can only establish communication link with the covered satellite in the t−th time interval, we denote the set of association candidate satellite for $g_{k}$ as $\Phi_{s}$, thus we have
(21)$$\Phi_{s}=\left\{v_{i}^{t}|\omega \left(v_{i}^{t},v_{j}^{t}\right)=1,\forall v_{i}^{t}\in V_{s},v_{j}^{t}\in V_{g}\right\}.$$

For the ISL selection problem of different data flows $UT_{k}$, the satellite routing selection problem can be presented as
(22)$$\begin{array}{c} \begin{array}{cc} \min_{\omega } & \sum_{v_{i}^{t}\in \Phi_{s}}(\frac{d(v_{i}^{t},v_{j}^{t})}{v_{P}}+\frac{q_{i}^{t}\left(k\right)}{r(v_{i}^{t},v_{j}^{t})}+\frac{\vartheta (v_{i}^{t},v_{j}^{t},k)}{r(v_{i}^{t},v_{j}^{t})}) \end{array}\omega \left(v_{i}^{t},v_{j}^{t}\right)\\ \begin{array}{ccc} s.t & C1,C4,C5. & \end{array} \end{array}$$

This optimization problem is equivalent to determining the shortest path between satellites, based on the dynamic link information, a Time interval optimization-based Satellite Routing (TSR) algorithm is proposed to solve the shortest path problem, which is summarized in Algorithm 1. Firstly, TSR checks that if the destination satellite $v_{d}$ is an alive neighbor node of source satellite $v_{0}$ in the initialization phase. If $v_{d}^{t}\in N_{0}$, the TSR algorithm will be ended and the data packet can be transmitted to the destination $v_{d}$ directly. If $v_{d}^{t}\notin N_{0}$, the corresponding satellites of all alive neighbor satellites in $v_{0}$ will be added into source set $\Phi_{0}$. The transmission rate from $v_{i}$ to the alive neighbor node is used to calculate the transmission delay. Algorithm 2 is in charge of maintaining the link delay, and the greedy approach is used to find the shortest path from source satellite $v_{0}$ to destination satellite $v_{d}$.
**Algorithm 1** Time Interval Optimization-based Satellite RoutingInput: Space-Time graph $G_{s}\left(V_{s},E_{ss}\right)$, source satellite $v_{0}\in \Phi_{s}$, TTL of data packet, start time *t*, data packet size $x(v_{i}^{t},v_{j}^{t},k)$Output: Router Table $R_{s}$1:**Initialize:**   Update router table and searched set $\Phi_{d}$ by executing Algorithm 1   Let source set $\Phi_{0}=\left\{s_{0},N_{s}\right\}$   Set termination time $t_{n}=t+TTL$   Optimal set $V_{opt}=\left\{T\left(v_{i}^{t}\right),p\left(v_{i}^{t}\right)|v_{i}^{t}\in V_{s}\right\}=\emptyset$2: **If** $N_{0}==\emptyset$ **then**3:    Return $R_{s}$4: **End**5: **While** $\left(V_{opt}\neq \emptyset \right)$ **do**6:    **foreach** $v_{i}^{t}\in \Phi_{0}$ **do**7:       **If** $v_{i}^{t}\in \Phi_{d}$ **then**8:          Continue loop in line 69:       **End**10:    $\Phi_{d}=\Phi_{d}\cup \left\{v_{i}^{t}\right\}$11:    Neighbor$(v_{0}^{t})\leftarrow$ Find all non-empty neighbor set of $v_{i}^{t}$12:    Update $V_{opt}$ by executing Algorithm 113:    **End for**14:    Extract $v_{\min}^{t}$ which has minimum delay $T_{\min}$ from $V_{opt}$15:    Add $v_{\min}^{t}$ into $\Phi_{0}$16:    Add $\left\{T_{\min},p_{\min}\right\}$ into $R_{s}$17: **Return** $R_{s}$

**Algorithm 2** Link Delay Maintenance
Input: Source satellite $v_{0}\in \Phi_{s}$, start time *t*, terminal time $t_{n}=t+\Delta \tau$, data packet size $\vartheta (v_{i}^{t},v_{j}^{t},k)$, neighbor set $Neighbor(v_{i}^{t})$
Output: The optimal set $V_{opt}$
1:**Foreach** $v_{j}^{t}\in Neighbor\left(v_{i}^{t}\right)$
2:    **If** $v_{j}^{t}\in \Phi_{0}$ **then**
3:       Continue loop in line 1
4:    **End**
5:    **If** $T\left(v_{j}^{t}\right)\geq t_{n}$
6:       Continue loop in line 1
7:    **End**
8:    $T\left(v_{i}^{t}\right)=T\left(v_{j}^{t}\right)+T\left(v_{i}^{t},v_{j}^{t}\right)$
9:    **If** $T\left(v_{i}^{t}\right)\leq V_{opt}\left(v_{i}^{t}\right)$ **then**
10:       $p\left(v_{i}^{t}\right)=p\left(v_{j}^{t}\right)$ , put $E(v_{i}^{t},v_{j}^{t})$ into $p(v_{i}^{t})$;
11:       Replace $V_{opt}\left(v_{i}^{t}\right)$ with $\left\{T\left(v_{i}^{t}\right),p\left(v_{i}^{t}\right)\right\}$ ;
12:    **End**
13: **End**
14: **Return** $V_{opt}$


#### 4.2.2. Hybrid Routing Algorithm Based on Bipartite Graph

Since TSE may choose different candidate satellites as the access satellite for transmitting the k-th data flow, based on the TSR algorithm, the end-to-end hybrid routing problem in TSIN can be constructed as follows
(23)$$\begin{array}{c} \begin{array}{cc} \min_{\omega ,\vartheta } & \sum_{\left(s_{i}^{t},s_{j}^{t}\right)\in \Phi_{s}}[T\left(v_{i}^{t},v_{j}^{t}\right)\omega \left(v_{i}^{t},v_{j}^{t}\right)+T_{\min}\left(s_{i}^{t},s_{j}^{t}\right)], \end{array}\\ \begin{array}{cc} s.t & C2-C6. \end{array} \end{array}$$

The optimization problem can be regarded as a matching problem in the bipartite graph. To solve this problem, the link connection during each time interval t(1≤t≤D) is modeled as a weighted bipartite graph $G^{{}^{\prime }}\left(V_{1},V_{2},E^{{}^{\prime }}\right)$, wherein, $V_{1}$ represents the set of candidate satellites associated with TSE, $V_{1}\in \Phi_{s}$. $E^{{}^{\prime }}$ represents the path set between candidate satellites and the link set between UTs and candidate satellites, the weight set of $E^{{}^{\prime }}$ is represented as follows
(24)$$\delta \left(v_{i}^{t},v_{j}^{t}\right)=\left\{\begin{array}{c} T_{\min}\left(v_{i}^{t},v_{j}^{t}\right),\left(v_{i}^{t},v_{j}^{t}\right)\in \Phi_{s},\\ T(v_{i}^{t},v_{j}^{t}),otherwise. \end{array}\right.$$

Let $c(UT_{k}^{t})$ denote the data volume of $UT_{k}$ through link $\left(s_{i}^{t},s_{j}^{t}\right)$, then the data volume in the next interval is expressed as:(25)$$\vartheta \left(v_{i}^{t},v_{j}^{t},k\right)=\vartheta \left(v_{i}^{t-1},v_{j}^{t-1},k\right)-c\left(UT_{k}^{t}\right).$$

Based on the constructed bipartite graph, minimum weight matching is modeled to obtain the link connection planning between UTs. Since the Kuhn–Munkres (KM) algorithm is an efficient matching algorithm [40], KM algorithm is used to solve the optimal matching problem on the bipartite graph. Algorithm 3 gives the KM-based link selection algorithm to obtain the best matching of $G^{{}^{\prime }}\left(t\right)$, then the shortest delay hybrid path can be found through Algorithm 4.
**Algorithm 3** KM-based Link SelectionInput: Bipartite graph, the minimum matching H(t) of $G^{0}\left(t\right)$Output: link selection in t-th $R_{t}$1:**Initialize:** $l(v_{i}^{t})$2: **Foreach** $G_{l}\left(t\right)\in G^{{}^{\prime }}\left(t\right)$ **do**3:    **If** $H\left(t\right)\notin R_{t}$ **then**4:       $P=\{v_{j}^{t}\},Q=\emptyset$, $N_{p}=G^{{}^{\prime }}\left(t\right)\cup \left\{P\right\}$5:       **If** $N_{p}\neq Q$6:          Continue loop in line 27:       **End**8:       **If** $N_{p}=Q$ **then**9:       $\Delta =\min_{u,v}\left\{l\left(v_{i}^{t}\right)|l\left(v_{i}^{t}\right)+l\left(v_{j}^{t}\right)-w\left(v_{i}^{t},v_{j}^{t}\right),v_{i}^{t}\in P,v_{j}^{t}\in V_{2}-Q\right\}$10:       $l^{{}^{\prime }}\left(v_{i}^{t}\right)=\left\{\begin{array}{c} l\left(v_{i}^{t}\right)-\Delta ,v_{i}^{t}\in P\\ l\left(v_{i}^{t}\right)+\Delta ,v_{i}^{t}\in Q\\ l(v_{i}^{t}),otherwise \end{array}\right.$11:       **End**12:    **End**13:    Extract H(t) in R(t)14: **End**15: **Return** $R_{t}$

In TSHR, the source TSEs will collect the link state and queuing delay information of each satellite to build the routing information table. To collect the real-time status of neighbor nodes, each TSE should periodically broadcast data packets to neighbor satellites and terrestrial nodes. After receiving the response packet, the neighboring relationship is established and link information would be recorded in the routing data table. As shown in Figure 5, for the node’s RREP header, the bandwidth and cache packet information would be writen in the reserved domain, then the information is exchanged with their neighbors by the RRER packet. In the beacon protocol, each satellite node will broadcast the beacon packet periodically to collect status (e.g., queuing delay, transmission rate) of neighbor satellites. When the beacon packet is received, the sender will be marked as an alive neighbor and information contained in the beacon packet will be saved. When there are no packets received after three broadcast periods, the link will be regarded as interrupted and the neighbor will no longer be treated as alive. In the following figure, satellite-1, satellite-2, and satellite-3 are neighbors with each other. Since satellite-3 can receive a beacon packet from satellite-1 and satellite-2, they will be marked as alive neighbors of satellite-3. The link between satellite-1 and satellite-3 is interrupted, thus satellite-1 will be deleted from the alive neighbor list of satellite-3. In this way, the connectivity of adjacent satellites can be gathered by TSEs, and will not increase the signaling overhead.
**Algorithm 4** KM-based Link-state Aware Terrestrial-Satellite Hybrid Routing algorithmInput:Network topology;Output: Link planning of hybrid routing $R_{h}$1:**Initialize:**   Update $R_{s}$ by executing Algorithm 22: **If** t≤D **then**3:    Construct bipartite graph of $G^{{}^{\prime }}\left(t\right)$4:    Calculate link metric of $G^{{}^{\prime }}\left(t\right)$ according to (Equation 24)5:    Calculate the minimum weight match of $G^{{}^{\prime }}\left(t\right)$ by using Algorithm3,   Extract hybrid path planning in t-th R(t)6:    Write in link connection planning $R_{h}$7:    Update data volume of link according (Equation 25)8:    t←t+19: **End**10: **Return** $R_{h}$

### 4.3. Complexity Analysis

In this subsection, we analyze the complexity of the proposed algorithm. Let *N* denote the number of candidate satellites associated with TSE in the space–time graph, the total number of executions about the loop in line 5 of Algorithm 2 equals *N*. To update the optimal set $V_{opt}$ in TSR, the maintenance cost for each loop is O(N), and the total number of calculations about cumulative transmission time in line 8 of Algorithm 1 is $E_{ss}$. Consequently, the computation complexity of TSR is $O(N^{2}+E_{ss})$. The computation complexity of solving the matching problem is $O(D.|v_{g}{|}^{3})$, then the total computational complexity is $O\left(N^{2}+E_{ss}\right)+O\left(D.|v_{g}{|}^{3}\right)$.

## 5. Simulation and Analysis

In order to evaluate the routing performance of the proposed algorithm, a Hongyun constellation is created in the Satellite Tool Kit (STK) [41], which can provide realistic 2D and 3D visual dynamic scenes of the constellation orbit. The performance of the space–time Topology-based Satellite Routing (TSR) algorithm and Terrestrial-Satellite integrated network Hybrid Routing (TSHR) algorithm are respectively carried out in the QualNet simulator by using real-world satellite parameters supplied from STK. Meanwhile, packet delay, throughput and packet drop rate are adopted as the performance metrics. In the experiments, the proposed TSR algorithm is compared with the Dijkstra Shortest-Path (DSP) [23], Traffic-Light-based Routing (TLR) [25], Netgrid-based Shortest path Routing (NSR) [21] and Selective Split Load Balancing routing Strategy (SSLB) [26]. Specifically, DSP adopts the gathered global propagation delay to find the optimal routing path, TLR is a load balancing routing scheme, which considers local route adjustment, NSR utilizes static cubes to find the shortest distance path, and SSLB is a distributed strategy, which diverts the traffic flow of congestion nodes to neighboring nodes. The proposed TSHR method is compared with the improved Open Shortest Path First+ (OSPF+) algorithm [17], Traffic-Aware Contact plan (TACR) algorithm [36], and Hyperbolic Geometry-based Routing (HGR) algorithm [34]. Specifically, OSPF+ takes advantage of the regularity of the constellation and conducts periodic route calculations to find the shortest distance path, TACR integrates traffic information into a mixed integer linear programming to find the optimal link connection planning, and HGR utilizes greedy forwarding strategy and hyperbolic geometry coordinate mapping to achieve optimal path selection. We repeat the same simulation 10 times with different generated packets for each point in routing performance comparisons, and we plot the average value with 95% confidence interval in all figures.

### 5.1. Simulation Setup

As shown in Figure 6, the satellite simulations are conducted based on a realistic Hongyun constellation project, the parameters of satellite constellation are presented in Table 4. For the delay-sensitive communication session, an acceptable delay variation of ISL length over the time interval is about 3 ms. Meanwhile, according to [20] and Hongyun constellation parameters, the upper bound number of interplane ISLs is 12, then Δτ is equal to 51 s by using (Equation 20). The real-flow in the network is self-similar and heavy-tail distributed, which shows flow’s statistical consistency under different time scales, and the On-Off model can explain the generation process of self-similar flow and decompose the data traffic into multiple source-destination pair flows [42]. Thus, we consider 600 non-persistent On/Off flows. The On/Off periods of the connections are derived from a Pareto distribution with a shape equal to 1.2. The average burst time and the average idle time are set to 200 ms. The source and destination terminals are dispersed all over the Earth, divided into six continental regions, following a distribution identical to the traffic distribution used in [43], which taking into account the geographic distribution of sources and destinations. The source terminals send data at varying from 0.8 Mbps to 1.5 Mbps.

To verify the routing performance when occurring network congestion, background flow and burst flow are set in the simulation. As shown in Figure 6, background flow refers to the original traffic on the ITL, i.e., the connected black line from $g_{m}$ to $g_{l}$ (∀m,l∈[1,10],m≠l ). Burst flow represents the sudden increase of traffic rate on the passing ITL, in the experiment, we set the burst flows from $g_{6}$ to $g_{7}$, i.e., the connected red line, the transmission rate of each burst flow is 5 Mbps, obeying an exponential distribution with a mean 100 ms.

### 5.2. Satellite Routing Performance

#### 5.2.1. Transmission Delay

Since the congesting satellite may lead to significant queuing delay, in the TSR method, packets would be transferred with more hops than traditional routing algorithms to alleviate congestion and guarantee low communication delay. To verify the effectiveness of the TSR method, the average end-to-end delay is presented in Figure 7. The delay of DSP is the highest among all simulated methods, which is mainly because it always transmit data through the shortest path and long queuing delay begins to appear at the selected path with heavy loads. In SSLB, the traffic of the congestion node is partially diverted to the neighbor node, which may aggravate the load of the neighbor node and cause new link congestion. As a result, some packets may aggregate at some satellites and finally increase the end-to-end delay. In the NSR method, the shortest path is estimated according to the transmission delay, then each intermediate node could adjust the routing path according to the real-time status of neighbor satellites. With the increase of congestion nodes, the number of re-routing process is rising, so that the end-to-end delay of NSR also rises. Since TLR dynamically detours traffic to alternative paths in response to congestion, the delay is substantially lower than the DSP, SSLB, and NSR. However, as the propagation delay of TLR is constant, the packets may be sent through a route with high propagation delay, so the end-to-end delay is higher than the TSR method. To this end, the TSR is proven to be effective in alleviating traffic congestion and delay reduction.

To further investigate the congestion control performance, the average queuing delay is plotted in Figure 8. In the simulation, the source UT sends data with the rate of 1.5 Mbps. The results indicate that the DSP algorithm gains a high queuing delay at some satellites, because it only considers the propagation delay when choosing the route, and some packets may aggregate at specific satellites. Since NSR and SSLB can adjust routing paths according to the real-time transmission delay of neighbor satellites, the average queuing delay of NSR and SSLB is substantially lower than the DSP. The queuing delay is calculated in TLR, thus the peak points become lower. However, due to the dynamic topology, a certain ISL in the middle is disconnected. In TLR, packets can only wait for link reconnection or TTL timeout to be discarded at the intermediate node, resulting in higher queuing delay. In the proposed TSR method, the dynamic satellite topology is represented by using the space–time graph model, and the path with the minimum delay is obtained, which can alleviate congestion. Thus, the peak values of queuing delay decline remarkably.

#### 5.2.2. Packet Drop Rate and Network Throughput

Apart from data transmission delay, the packet drop rate and network throughput are also crucial metrics for network performance analysis, thus we further adopt them to verify the routing effectiveness. Figure 9 shows the total packet drop rate experienced by all sessions for different sending rates. Obviously, TSR achieves the least packet drop rates among all simulated methods. Since the packet data rate is mainly caused by traffic congestion, thus the performances of these four simulated methods are similar to the end-to-end and queuing delay as shown above. The more data choose the same way for transmission, the higher the packet drop rate. The good performance of the TSR method in the lower packet drop rate is also reflected in the high throughput. As shown in Figure 10, the proposed TSR algorithm achieves a better performance in the throughput, the reason is that the TSR algorithm uses the space-time graph model to depict the data transmission process in dynamic TSIN, which avoids the data packet drop caused by ISL disconnection, and the inter-satellite link connection with the minimum delay is obtained, which can achieve load balancing.

### 5.3. Terrestrial–Satellite Routing Performance

Figure 11 and Figure 12 show the comparison results of transmission delay and throughput between the proposed TSHR, OSPF+, TACR, and HGR, respectively. Obviously, the TSHR outperforms OSPF+, TACR, and HGR in terms of end-to-end transmission delay and throughput. Since the proposed TSR algorithm can dynamically detour traffic to alternative paths in response to congestion, the throughput is substantially higher than the TACR and OSPF+. Although TACR utilizes the topology information provided by the contact graph, it searches future available neighbor nodes. Considering the time-varying network topology, the path planning might be sub-optimal or even invalid. As the background flow increases to the range 5–7 Mbits, the non-congested alternative paths reduce. Therefore, the throughput of the TSHR algorithm starts to increase slowly, and the performance gap between the TSHR and TACR is the minimum. However, as the background flow increases to 11 Mbits, TACR always transmits data through the shortest path and a long queuing delay begins to appear at the selected path with heavy loads, which results in rapid throughput degradation. Thus, the performance gap between the TSHR and TACR increase as shown in Figure 12. HGR utilizes greedy forwarding strategy and hyperbolic geometry coordinates mapping to achieve path selection, which fails to consider the congestion; therefore, some packets are discarded before they are sent out. Since the routing table is calculated independently in each satellite in OSPF+, the large route calculation interval will cause severe delay and throughput degradation. Furthermore, the delay grows nonlinearly with the route calculation interval due to the discreteness of the constellation topology change. On the contrary, to avoid high queuing delay, alternative paths are calculated in the proposed TSHR when terrestrial network is about to get congested, the dynamic satellite topology is represented by using the space–time model, then the TSL and ISL with the minimum delay is obtained through bipartite graph mismatch. To sum up, the transmission in the satellite network helps to alleviate the traffic burden and leads to better delay and throughput performance.

Figure 13 shows the comparison results of transmission delay in network congestion situation and the proposed TSHR. At the beginning of transmission, since there are enough network bandwidths to serve the requested traffic of UTs, the selected path by TSHR is the same as the shortest path of terrestrial routing. At 150 s, to simulate the network congestion, background flows or burst flows are triggered, the TSHR significantly outperforms the terrestrial routing algorithm. The reason is that the terrestrial routing algorithm only finds the shortest path, data is then transmitted over the shortest paths during the entire communication session; hence, there would be a large delay and low throughput when some ITLs are congested. On the contrary, to avoid high queuing delay, an alternative paths is calculated in TSHR when the terrestrial network is about to get congested, which means that the data are transmitted via the terrestrial-satellite hybrid path, namely source UT-TSE-satellite network-TSE-destination UT. To sum up, the transmission in the satellite network helps to alleviate the traffic burden and leads to better delay performance.

## 6. Conclusions

In this paper, to provide seamless integration of the terrestrial network and satellite network, a link-state aware terrestrial–satellite integrated network hybrid routing algorithm is proposed for data transmission in the dynamic TSIN. To represent the data transmission process, a space–time topology model is proposed. The link resources in both the spatial and temporal dimensions are quantified by dividing them into discrete time intervals. Then, based on the space-time topology, with the objective of minimizing the end-to-end delay, the routing problem of link selection is formulated as an integer linear programming problem. To obtain the data transmission path with the minimum delay, we propose the relationship function between the time interval and the satellite constellation parameters, and further propose a hybrid routing algorithm based on the bipartite graph model. Simulation results show that the proposed hybrid routing algorithm can achieve the best performance in terms of end-to-end delay, packet loss rate, and throughput.

It is worthwhile to mention that the introduced concept of hybrid routing and the space–time topology graph is possible to be extended into other relevant satellite link connection design problems, e.g., in satellite-enabled wireless collaborative transmitting, mission scheduling of mobile edge computing, and sensing. In addition, for future 6G heterogeneous networks, UAVs/HAPs would communicate with ground and satellite stations, forming a space–air–ground network facing frequent satellite–UAV/HAPs–ground link switching. The dynamic 6G heterogeneous networks can be represented by the proposed space–time topology graph, in which the collaborative communication with each other can be described via the edges between them. During the design of the topology model from the transmission period, we derive an optimal time interval approach for the topology relationship, which has a high potential in addressing other satellite link selection problems. Moreover, the methodology of this work in handling the time-varying and limited inter-satellite link capacity can be extended to carry out more designs for TSINs. For our future work, we will explore the hybrid routing for serving the communication for massive Internet-of-Thing equipment. In order to provide ubiquitous and saleable networks, we also further explore UAV/HAPs mobility to constuct the graph-based routing approach in the 6G heterogeneous network.

## Figures and Tables

**Figure 1 sensors-22-09124-f001:**
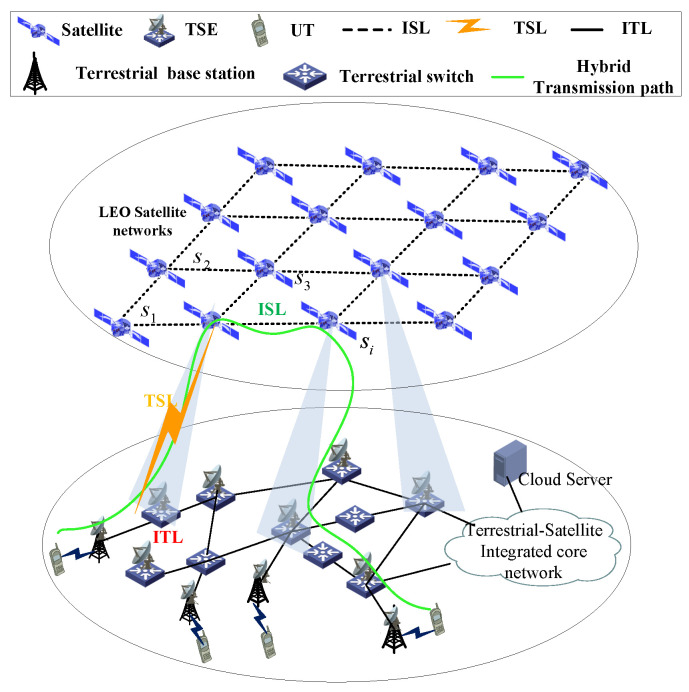
The transmission architecture of TSIN.

**Figure 2 sensors-22-09124-f002:**
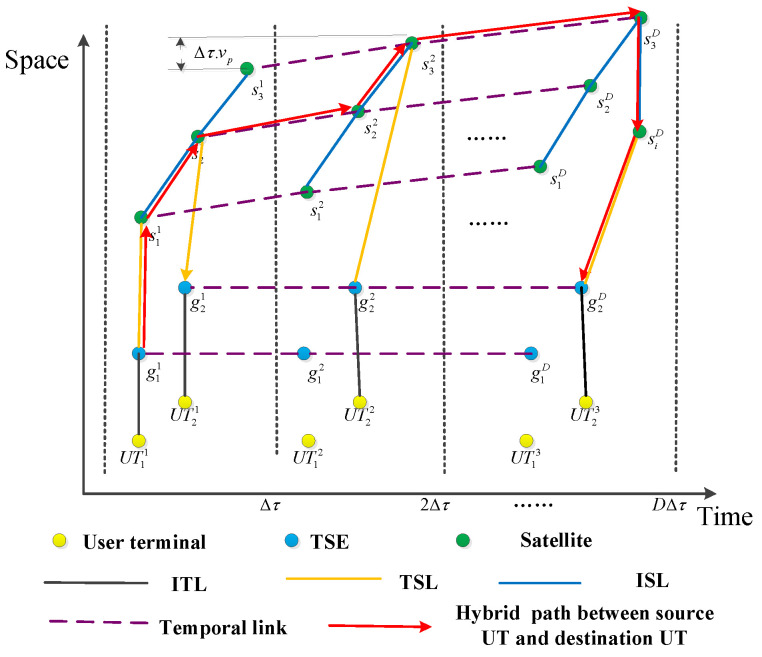
Space–time topology model.

**Figure 3 sensors-22-09124-f003:**
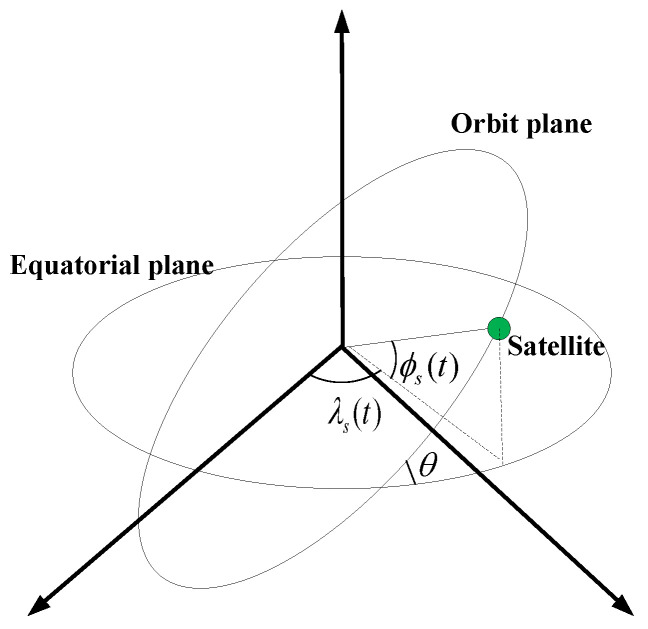
The geometry trajectory of satellite.

**Figure 4 sensors-22-09124-f004:**
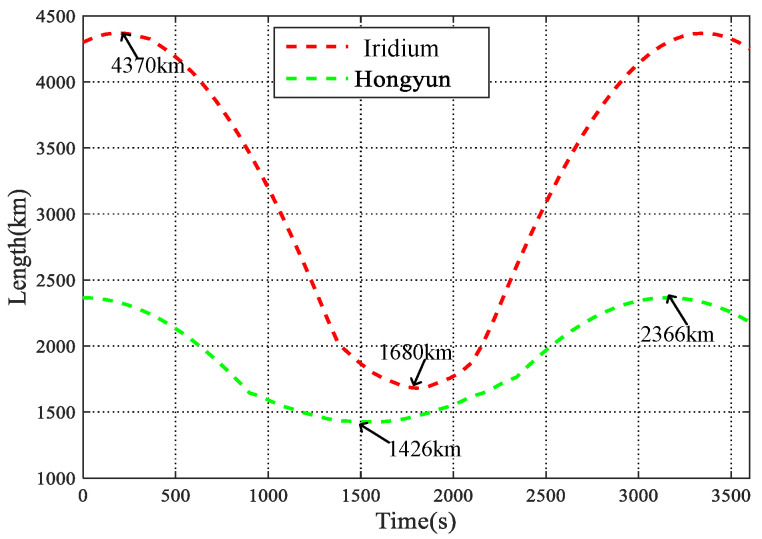
Interplane ISL distance vs. time.

**Figure 5 sensors-22-09124-f005:**
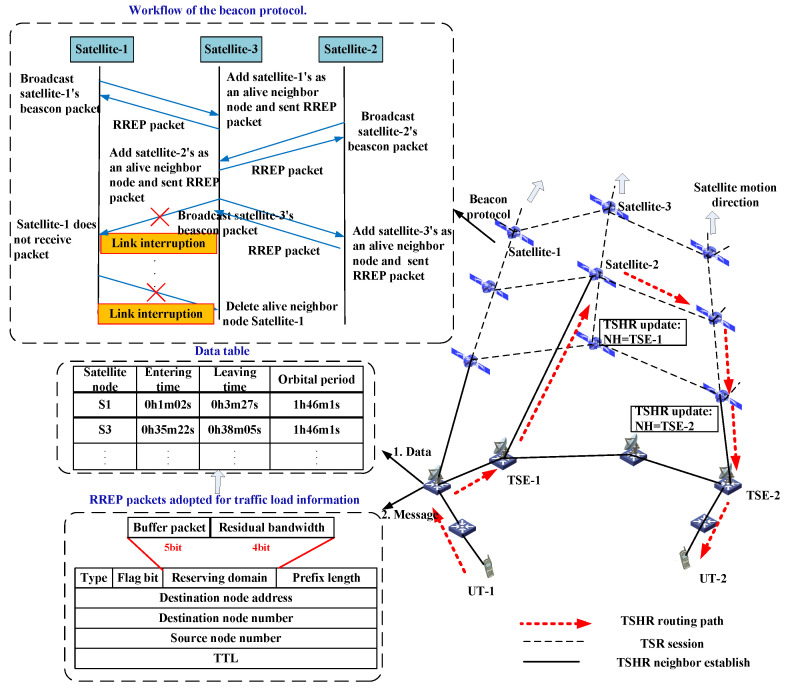
TSHR signaling workflow.

**Figure 6 sensors-22-09124-f006:**
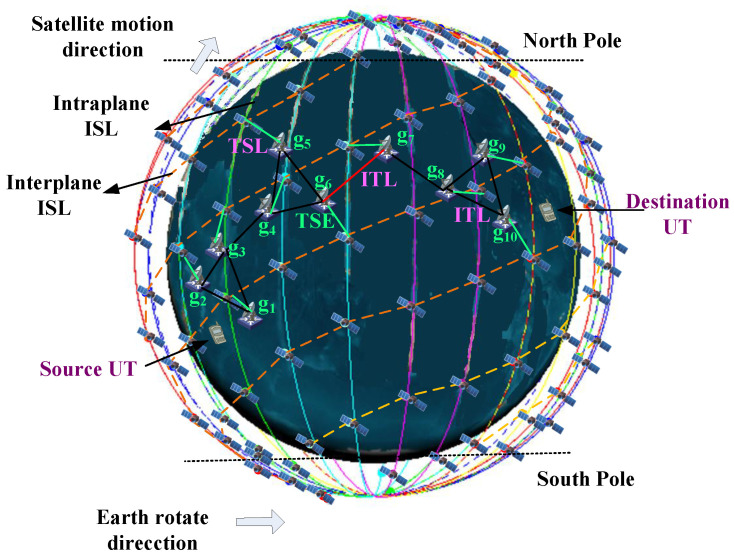
Simulation topology in a time interval.

**Figure 7 sensors-22-09124-f007:**
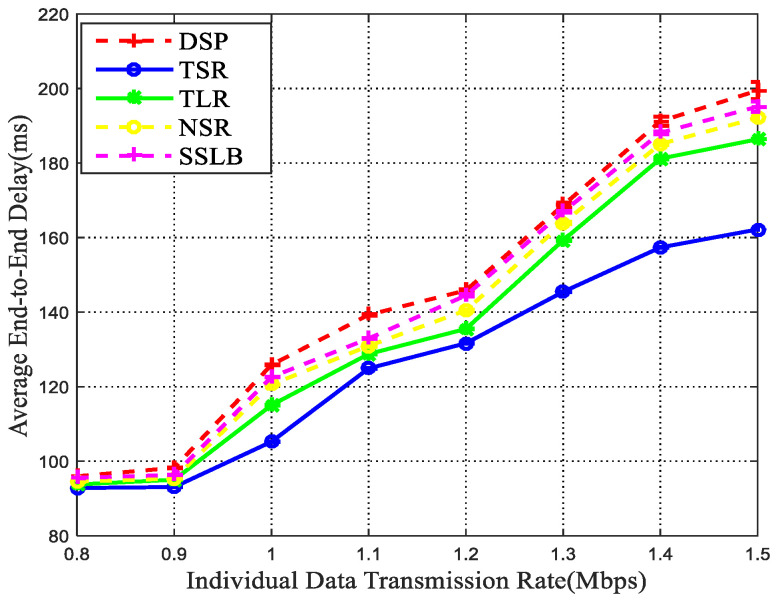
End-to-end delay vs. transmission rate.

**Figure 8 sensors-22-09124-f008:**
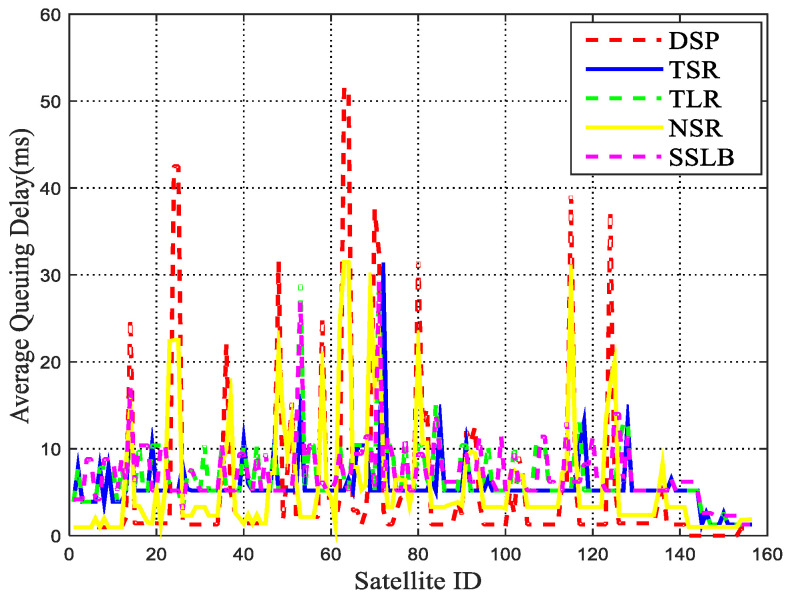
Average queuing delay at each satellite.

**Figure 9 sensors-22-09124-f009:**
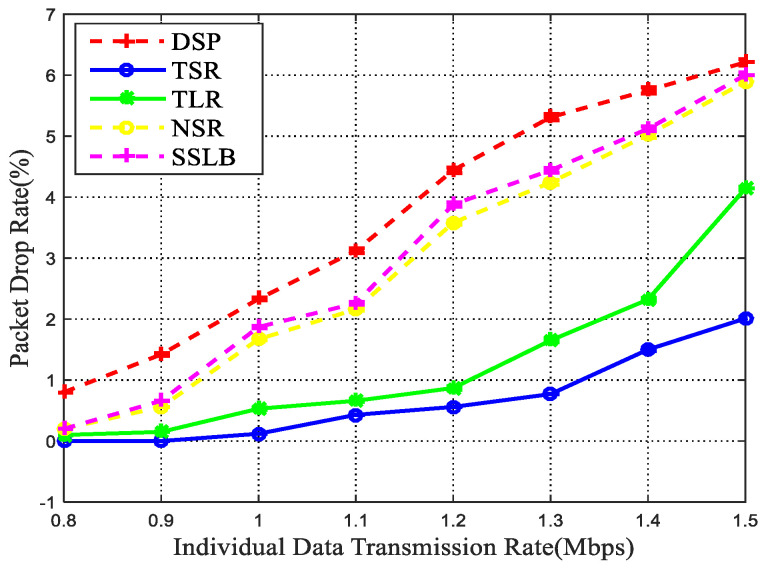
Packet drop rate vs. data transmission rate.

**Figure 10 sensors-22-09124-f010:**
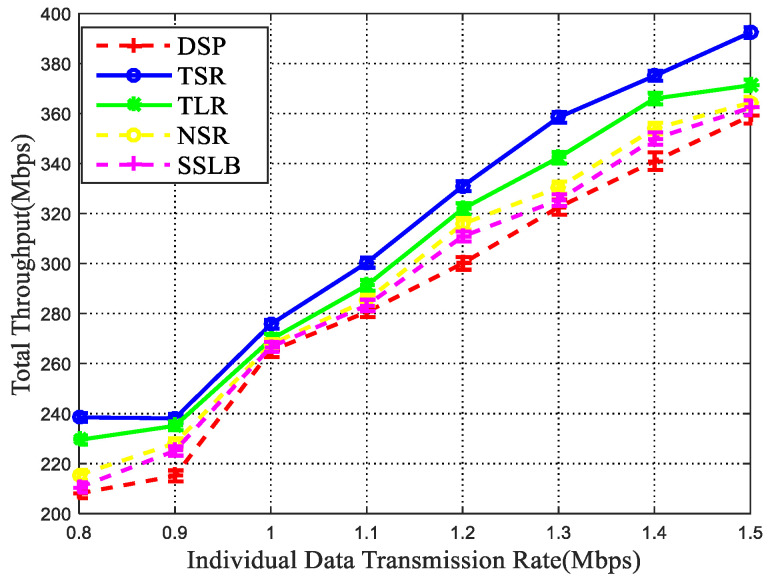
Network throughput vs. data transmission rate.

**Figure 11 sensors-22-09124-f011:**
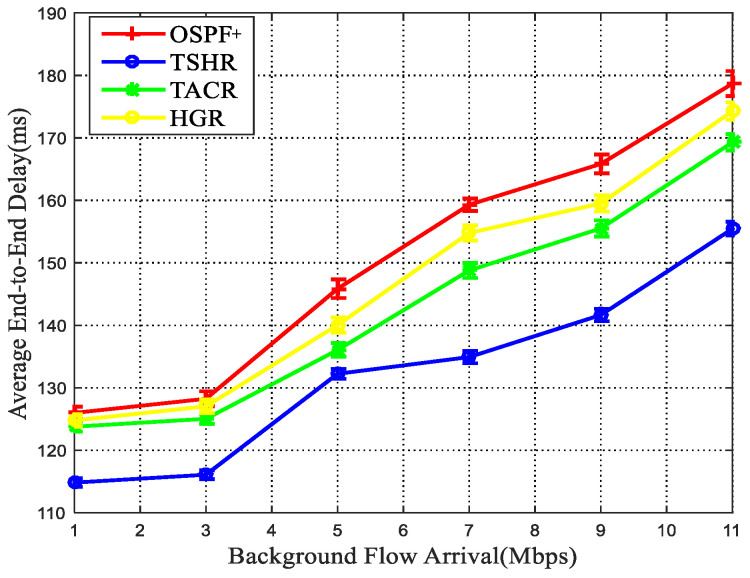
Delay vs. background flow arrival.

**Figure 12 sensors-22-09124-f012:**
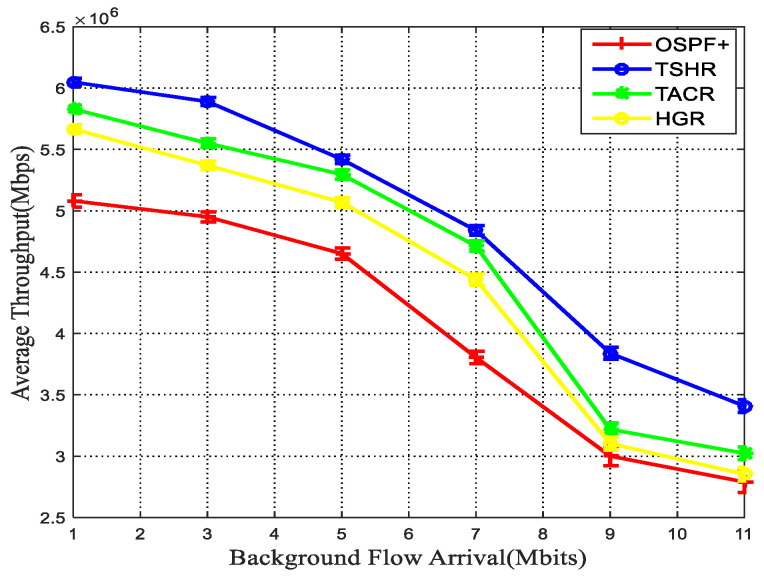
Throughput vs. background flow arrival.

**Figure 13 sensors-22-09124-f013:**
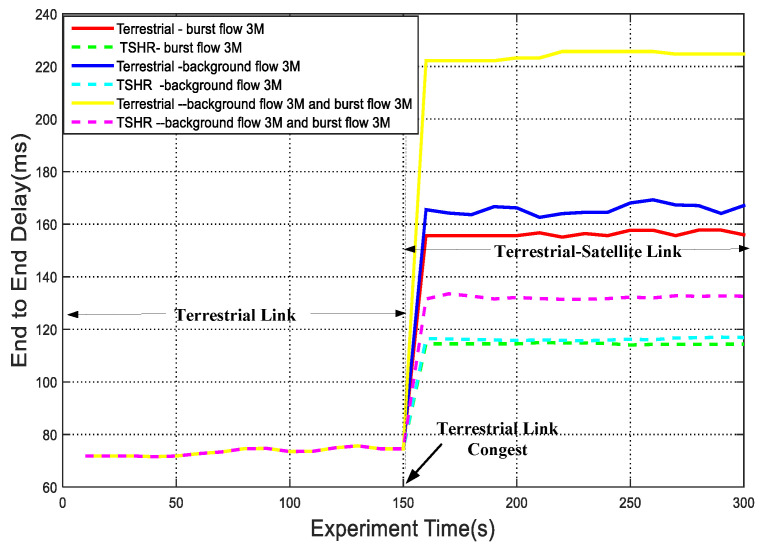
End-to-end delay in congestion situation.

**Table 1 sensors-22-09124-t001:** Summary of routing protocols in TSIN.

Routing Schemes		Core Idea	Limitations
Satellite routing schemes	Routing algorithm based on virtual topology [18,19]	The system period is divided into equal-length snapshots, and the route of each snapshot is calculated offline	The routing table needs to be frequently synchronized
	Routing algorithm based on virtual node [20,21]	Virtual nodes are set to represent physical satellites, and the shortest distance route between virtual nodes is calculated	The link congestion and failure would deteriorate throughput performance
	Routing algorithm based on load balancing [22,23,24,25,26]	Based on the traffic state information, the route is adjusted dynamically according to the real-time state at intermediate nodes to avoid congestion	The intermittent problem of link connection caused by dynamic topology has not been solved
Hybrid terrestrial -satellite routing schemes	Relay routing [13,14]	Satellites act as relay forwarding nodes, which are interconnected through ground stations	The scalability of satellite networks is limited
	Extended terrestrial routing [15,17]	Terrestrial routing algorithms are extended to the satellite networks	Dynamic topology leads to frequent path updates
	SDN-based routing [27,28,29]	The control center collects the link information of the whole network and calculates the route	The time-varying topology of satellites increases the computation overhead
	Graph-based routing [30,31,32,33,34,35,36,37,38]	Communication link planning algorithm is designed according to the scheduled contact graph	The frequent changes of inter-satellite link and terrestrial-satellite link selection is missing

**Table 2 sensors-22-09124-t002:** Notations and definitions.

Notations	Definitions
*S*	The set of satellites $s_{i}$
*G*	The set of TSEs $g_{i}$
UT	The set of UTs $UT_{i}$
*D*	The total number of time intervals in path plan
Δτ	The duration of each time interval
G(t)	The directed graph composed of T layers
*V*	The vertices set of *G*
E(t)	The links set of *G*
$E_{l}\left(t\right)$	The links set of spatial link
$E_{b}\left(t\right)$	The links set of temporal link
$d(v_{i}^{t},v_{j}^{t})$	The distance of link $\left(v_{i}^{t},v_{j}^{t}\right)$
$v_{P}$	The transmission speed of radio signal
$r(v_{i}^{t},v_{j}^{t})$	The capacity of link $\left(v_{i}^{t},v_{j}^{t}\right)$
$b_{i}^{t}\left(k\right)$	The data volume in storage of $v_{i}^{}$ for $UT_{k}$ during the t-th time interval
$\vartheta (v_{i}^{t},v_{j}^{t},k)$	The capacity of link $\left(v_{i}^{t},v_{j}^{t}\right)$ for $UT_{k}$ data
$\omega (v_{i}^{t},v_{j}^{t})$	The connection state of link $\left(v_{i}^{t},v_{i}^{t}\right)$
$T_{p}\left(v_{i}^{t},v_{j}^{t}\right)$	The propagation delay
$T_{q}\left(v_{i}^{t},v_{j}^{t}\right)$	The queuing delay
$T_{w}\left(v_{i}^{t},v_{j}^{t}\right)$	The transmission delay
$E_{gu}\left(t\right)$	The communication opportunities for UTs to TSE in each time interval
$E_{sg}\left(t\right)$	The communication opportunities for satellite to TSE in each time interval
$E_{ss}\left(t\right)$	The communication opportunities for satellite to satellite in each time interval
$C(v_{i}^{t},v_{j}^{t})$	The maximum capacity of $\left(v_{i}^{t},v_{j}^{t}\right)$
$r_{\min}$	The minimum rate requirement of data transmission

**Table 3 sensors-22-09124-t003:** The comparison between ISLs.

Constellation	Iridium	Hongyun
No. of planes	6	13
No. of satellites per plane	11	12
Orbital inclination	$86^{\circ }$	$86.5^{\circ }$
Orbital height	788 km	1000 km
Intraplane ISLs length	4032.9 km	3790 km
Intraplane ISLs propagation delay	13.5 ms	12.6 ms
Longest interplane ISLs distance	4370 km	2366 km
Longest interplane ISLs propagation delay	14.5 ms	7.89 ms
Shortest interplane ISLs distance	1680 km	1426 km
Shortest interplane ISLs propagation delay	5.6 ms	4.75 ms

**Table 4 sensors-22-09124-t004:** Simulation parameters.

Parameters	Values
Orbital altitude	1000 km
θ	87.4${}^{\circ }$
*N*	13
*M*	12
Minimum elevation angle	10${}^{\circ }$
Phase offset between interplane satellites	360/12/2 = 15${}^{\circ }$
Bandwidth of ISL	15 Mbps
Bandwidth of ground link	15 Mbps
Queue type	FIFO
Buffer queue size	100 packets
Packet sizes	1500 byte

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
