# Peer review of "Link-State Aware Hybrid Routing in the Terrestrial–Satellite Integrated Networkâ€"

_sensors, 2022, doi:10.3390/s22239124_

Round 1
Reviewer 1 Report
Overall this is an excellent study which contributes an effective link-state aware hybrid routing scheme in the terrestrial-satellite integrated network. Some suggestions are as follows:
1. The text is beyond the table border in Table 1 in Page 4.
2. Some typos still exist and a full English proofreading is suggested, e.g., in "Since there are depends on limited number of ground stations" in line 151 Page 5, "depends" cannot be used as a noun and "interval in all Figure" in line 427 Page 17 should be "interval in all figures".
3. A promising trend is to apply graph-based deep learning for a series of problems in communication networks, e.g., routing and resource allocation. It would be a future research direction since this study also uses the graph structure. More discussion can be added in the related work with the following references:
[1] Ivanov A, Tonchev K, Poulkov V, et al. Graph-Based Resource Allocation for Integrated Space and Terrestrial Communications. Sensors, 2022, 22(15): 5778.
[2] Jiang W. Graph-based deep learning for communication networks: A survey. Computer Communications, 2022, 185: 40-54.
[3] Liu M, Li J, Lu H. Routing in small satellite networks: A GNN-based learning approach. arXiv preprint arXiv:2108.08523, 2021.
[4] Wang H, Ran Y, Zhao L, et al. GRouting: Dynamic Routing for LEO Satellite Networks with Graph-based Deep Reinforcement Learning. 2021 4th International Conference on Hot Information-Centric Networking (HotICN). IEEE, 2021: 123-128.
4. The relationship between the proposed space-time Topology-based Satellite Routing (TSR) and Terrestrial-Satellite integrated network Hybrid Routing (TSHR) algorithms are not so clear. More relevant discussion should be added in the introduction section.
5. The legend overlaps with the curves in Figure 13 in Page 22 and a better plot is expected.
Reviewer 2 Report
In this study, the authors examine data transmission in the Terrestrial-Satellite Integrated Network (TSIN), where terrestrial networks and satellites are integrated to offer ground users with seamless global network services.
The authors proposed a link-state aware hybrid routing algorithm, which selects the integrated data transmission path adaptively.
The article is interesting and well-written. The article's methodology is correct.
-The authors must also address the technological aspect. The authors are requested to provide information regarding B5G and 6G networks and the effectiveness of the proposed method in these networks. Because terrestrial-satellite integration is the defining aspect of B5G networks. The authors may get idea from the article(https://ieeexplore.ieee.org/abstract/document/9915455).
-If feasible, Figure 1. should be redrawn (optional). The authors must clarify each entity and its operation in depth for the reader to have a complete understanding.
-The writers must expand on the future work. The authors must explore UAVs and HAPs equipment and concepts in future study.
- Overall, the article would be a valuable contribution to the scientific literature in the specific field.
Round 2
Reviewer 1 Report
Dear authors,
Thanks for revising and resubmitting the manuscript. No futher concerns.
Reviewer 2 Report
The topic is topical and interesting. The authors have satisfactorily addressed my comments, and I believe this work is suitable for publication.